# Rapid fabrication of hydrogel micropatterns by projection stereolithography for studying self-organized developmental patterning

Ye Zhu[1], Daniel Sazer[1], Jordan S. Miller[1]*, Aryeh Warmflash[1,2]*

**1** Department of Bioengineering, Rice University, Houston, Texas, United States of America, **2** Department of Biosciences, Rice University, Houston, Texas, United States of America

* jmil@rice.edu (JSM); aryeh.warmflash@rice.edu (AW)

**Data Availability Statement:** All relevant data are within the manuscript and its Supporting Information files.

**Funding:** This work was funded by Rice University and grants to AW from the Welch Foundation (C-

## Abstract

Self-organized patterning of mammalian embryonic stem cells on micropatterned surfaces has previously been established as an in vitro platform for early mammalian developmental studies, complimentary to *in vivo* studies. Traditional micropatterning methods, such as micro-contact printing (μCP), involve relatively complicated fabrication procedures, which restricts widespread adoption by biologists. Here, we demonstrate a rapid method of micropatterning by printing hydrogel micro-features onto a glass-bottomed culture vessel. The micro-features are printed using a projection stereolithography bioprinter yielding hydrogel structures that geometrically restrict the attachment of cells or proteins. Compared to traditional and physical photomasks, a digitally tunable virtual photomask is used in the projector to generate blue light patterns that enable rapid iteration with minimal cost and effort. We show that a protocol that makes use of this method together with LN521 coating, an extracellular matrix coating, creates a surface suitable for human embryonic stem cell (hESC) attachment and growth with minimal non-specific adhesion. We further demonstrate that self-patterning of hESCs following previously published gastrulation and ectodermal induction protocols achieves results comparable with those obtained with commercially available plates.

## Introduction

Self-organized differentiation of embryonic stem cells on micropatterned surfaces has emerged as a valuable method for studying signaling and cell-fate patterning in early mammalian development. In these protocols, ESCs are cultured on micropatterened substrates that geometrically confine cell colonies to be of a particular size and shape. Treatment of these colonies with appropriately chosen courses of growth factor stimulation recapitulates aspects of embryonic patterning at gastrulation and neurulation stages. In particular, several such systems have been developed and used to understand how dynamic morphogen signaling underlies patterning of germ layers at gastrulation or ectodermal cell fates at neurulation stages [1–10].

Several micropatterning techniques have been developed for use in these and other studies which require particular cell colony geometries. Micro-contact printing (μCP) is a widely used

2021), NSF (MCB-1553228), NIH (R01GM126122), and Simons Foundation (511079), and to DS and JM from the Robert J. Kleberg, Jr. and Helen C. Kleberg Foundation, and a training fellowship to DS from the Gulf Coast Consortia on the NSF IGERT: Neuroengineering from Cells to Systems (#1250104). The funders had no role in study design, data collection and analysis, decision to publish, or preparation of the manuscript.

**Competing interests:** The authors have declared that no competing interests exist.

method for micropatterning in cell biology (reviewed in [11]). In short, µCP involves template fabrication (usually by UV photolithography), polydimethylsiloxane (PDMS) stamp casting, and printing (i.e., transferring ink material onto culture substrate). Nevertheless, complex fabrication operations that require cleanroom usage restrict the usage of µCP. Photo-patterning is another classical micropatterning method, in which patterns are fabricated on a glass surface by spatially controlled exposure of photosensitive material to UV light ($\approx$ 200–385 nm) through a photomask with desired micro-features [12]. The photomask is either attached directly with the substrate [13] or placed in the focal plane of the objective of a microscope [14]. The time-consuming and expensive fabrication process of photomask lead to a more popular choice of commercialized photomask fabrication service by scientists. Nevertheless, if the photomask requires frequent design modification for experimental needs, the high expense and effort of photomask fabrication could be a hinderance.

Recently, Grigoryan, Paulsen and Corbett et al developed an open source stereolithography apparatus for tissue engineering (SLATE), which uses a digitally tunable virtual photomask projection and 405 nm blue light source to build three-dimensional tissue structures in a layer-by-layer process [15]. SLATE allows 3D printing of biocompatible hydrogels with multivascular structures and intravascular topologies embedded. Hydrogels with bioinspired alveoli structures were fabricated and used to study oxygenation of human red blood cells inside the hydrogel. SLATE was also used to build vascularized hydrogel carriers to deliver hepatic aggregates *in vivo*.

Here we show that this technique can also be used for two-dimensional (2D) micropatterning of hydrogel features on glass surfaces using a single virtual photomask. We present a simple and efficient method for fabrication of micropatterns on a glass surface for cell culture using SLATE, which (1) uses biocompatible materials (poly(ethylene glycol) diacrylate [PEGDA]) (2) avoids the need for cytotoxic UV wavelengths (3) can be applied directly on glass-bottomed petri dishes or multi-well slides, (4) is highly adaptable, allowing easy pattern changes with almost no cost by using virtual photomasks, (5) achieves spatially confined cell patterns on micropatterned culturing surface. The resulting patterns are suitable for micropatterned culture of human embryonic stem cells (hESCs) for research on self-organized developmental patterning or other applications.

## Materials and methods

### Methacrylation of glass surface

The protocol for methacrylation of glass surfaces was described in [16]. The goal of glass methacrylation is to covalently bind methacrylate functional groups to the glass bottom of the culture vessel. This allows the glass surface to participate in the photocrosslinking reaction and thus to covalently bond the PEGDA gel for longitudinal culture. Without methacrylation, the gel may detach from the glass surface within hours or days. Here, gels were directly fabricated on the surfaces of glass-bottomed dishes (Cellvis, California, USA) or 18-well µ-Slides (Ibidi GmbH, Gräfelfing, Germany). These vessels are pre-cleaned and sterilized for tissue culture by the supplier, and no further cleaning was needed before methacrylation. To methacrylate glass surfaces, a methacrylated silane coupling agent (3-(Trimethoxysilyl) propyl-methacrylate; TPM) was mixed with ethanol (1: 20) and dilute acetic acid (1:30). The dilute acetic acid was prepared by mixing acetic acid with MilliQ $H_2O$ (1: 10).

To each 35 mm glass bottom dish, 2 mL silane solution was added to cover the bottom; to each well of the 18 well µ-Slide, 150 µL silane solution was added. The filled dishes or slides were covered to reduce the evaporation rate of the silane solution and incubated for 12–24 hours in a chemical hood at room temperature. After incubation, the remaining silane solution

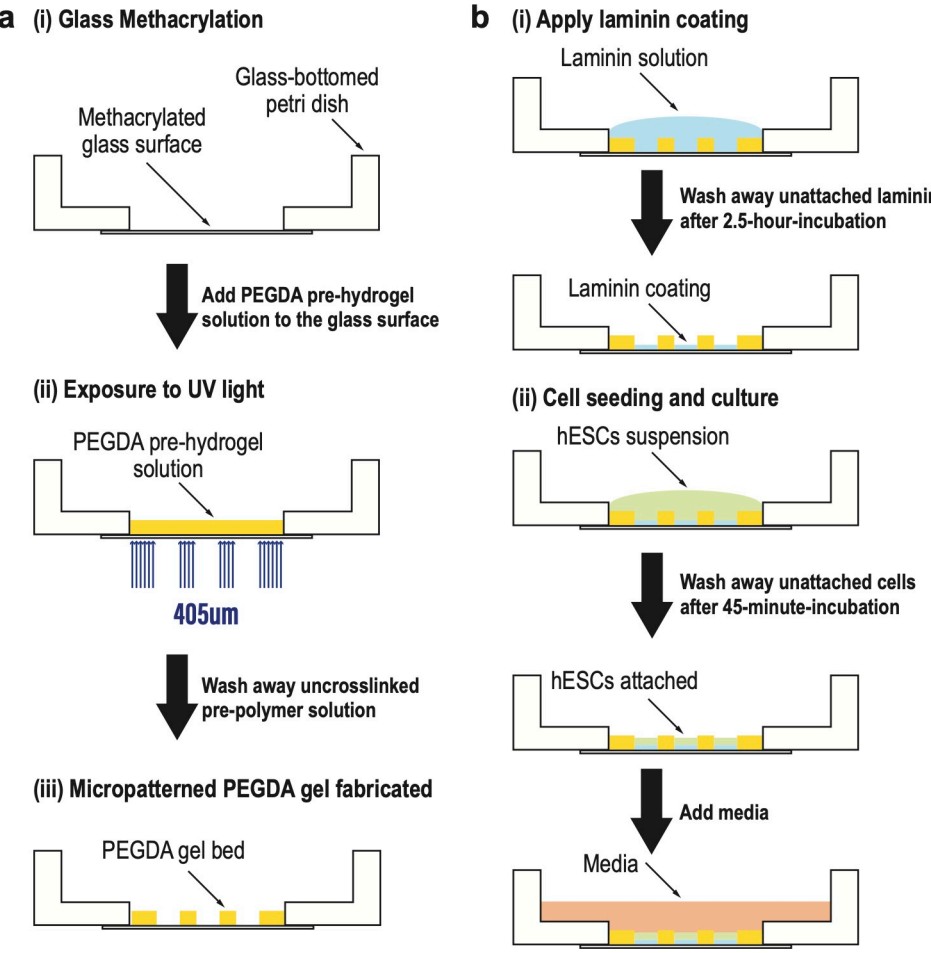

**Fig 1. The methodology for PEGDA-based micropatterning on methacrylated glass-bottomed petri dish by photo crosslinking.** (**a**) Schematic illustration of the fabrication process (i) Glass surface methacrylation; (ii) PEGDA pre-hydrogel solution was added onto the glass surface, and then exposed to UV light (wavelength = 405 μm) with the virtual photomask embedded; (iii) Micropatterned PEGDA gel was fabricated on the glass surface. Uncrosslinked pre-hydrogel solution was removed by multiple PBS washes. (**b**) Schematic illustration of hESC seeding and culture (i) Laminin coating was applied on the micropatterned surface. Unattached laminin was washed away after 2.5-hr-incubation at 37˚C; (ii) hESC suspension was added at desired density on the laminin coated surface. Unattached cells were washed away after 45-min-incubation at 37˚C. Following seeding, media with desired reagents was added to the dish for further culture or induction.

was removed. The dishes or slides were then rinsed with ethanol two times and sonicated in a 70% ethanol solution for 20 minutes. After air drying with $N_2$, the dishes or slides were baked at 60˚C for 5 hours. The methacrylated glass-bottomed dishes or slides were either used immediately or stored at 4˚C and protected from light (Fig 1A(i)).

## 2-Dimentional (2D) micropatterning fabrication of PEGDA hydrogels by projection stereolithography on glass-bottomed dish/slice

The stereolithography apparatus for tissue engineering (SLATE) was developed and described in [15]. The projection device consists of a PRO4500 Optical Engine (with 50 μm pixels and a display resolution of 1280x800 pixels) containing a 405 nm LED, attached to a computer for projection of virtual photomasks [15]. For micropatterning fabrication, the pre-hydrogel solution was prepared using 20 wt% 6 kDa PEGDA, 34 mM Lithium phenyl-2,4,6-trimethylbenzoylphosphinate (LAP)

as photo-initiator, and 2.25 mM tartrazine as a photo-absorber [15] to limit light scattering. Chemical synthesis of PEGDA and LAP was performed as previously described in [17] and [18], respectively.

The binary virtual photomasks were created using a custom MATLAB script, which generated standard black and white PNG images. The photomask image was then transferred to the SLATE computer. The culture vessel of interest (glass-bottom dish or μ-Slide) was first placed atop the transparent projection surface and filled with the desired volume of PEGDA pre-hydrogel solution. Photomasks were then projected directly through the glass surface into the photosensitive solution (Fig 1A(ii)), crosslinking the PEGDA polymer into a solid hydrogel in only the regions where incident light is received (Fig 1A(iii)). Crosslinking occurs within seconds, yielding a soft hydrogel micropattern on glass suitable for cell seeding and culture.

After irradiation, the un-crosslinked solution was washed away by multiple PBS washes and one 70% ethanol wash for sterilization. 1–2 mL of PBS was added into each dish, covering the entire micropatterned hydrogel, to avoid dehydration. The dishes or slides with micropatterned hydrogel were either used 1 hour after to allow swelling of the hydrogel or stored at 4°C and kept away from light (Fig 1A(iii)). The micropatterned dishes were still good for use after 8 months.

## Characterization of gel and pattern

**Measuring gel thickness.** In order to quantify the thickness of the transparent hydrogel micropatterns, polystyrene microspheres (1.0 μm), FluoSpheres, with yellow-green fluorescence were mixed at a ratio of 1: 200 (v/v) into the pre-hydrogel solution, taking care to avoid making bubbles. The hydrogels were then fabricated on the glass culture surfaces as described above using the pre-hydrogel solution with fluorescent microspheres, by using desired exposure times. After fabrication, the gels were washed with 1mL PBS 5 times to remove any remaining fluorescent microspheres.

Confocal images of the PEGDA gels with physically embedded fluorescent microspheres were acquired by using multi z-sections (thickness = 1 μm) with a 20X, NA 0.75 objective on an Olympus FV1200 laser scanning confocal microscope. z-sections were acquired from at least 10 μm below the bottom of the gel bed to at least 10 μm above the top of the gel bed. The acquired stack images were analyzed by using FIJI [19] to manually identify the top and the bottom of each gel, yielding a precise measurement of gel thickness.

**Measuring cell pattern size.** Images of cell colonies seeded on circular micropatterns of different sizes (diagonal length from 600 to 1000 μm) were taken at 10X resolution NA 0.25 on an Olympus CKX41 microscope, immediately after cell seeding. For each colony, a binary mask indicating the position of the colony was created by segmenting the image of the colony by using ilastik [20]. Then the masks were processed and analyzed using MATLAB scripts. The length of the major axis was taken as the size of the colony.

## Cell culture and differentiation

**Cell line and routine cell culture.** All cell attachment and patterning experiments were performed using ESI017 (obtained from ESI BIO, RRID: CVCL_B854, XX) hESC line or ESI017 NODAL knockout (NODAL -/-) cell line as described in [6]. All cells were grown in the chemically defined medium mTeSR1 in 35 mm cell culture dishes and kept at 37°C, 5% $CO_2$ as described in [21]. Cells were routinely passaged and checked for mycoplasma contamination also as described in [21]. In all experiments, cell passage number did not extend beyond 45.

**Cell seeding on micropatterned surface.** Micropatterned glass surfaces were coated with 5 μg/mL laminin-521 (LN521) in 1X PBS with calcium and magnesium (PBS++) for 2.5 hours at 37°C. The LN521 was first diluted 1:10 in PBS++ once, and all liquid was then removed

without touching the LN521 coated micropatterned glass surface (Fig 1B(i)). The LN521-coated dish or slide was either used immediately or stored for up to two weeks at 4°C with PBS++ covering the micropatterned surfaces.

Seeding of hESCs onto micropatterned surfaces was performed as described in [22]. hESCs were seeded onto LN521-coated micropatterned surfaces at $2 \times 10^6$ cells/mL, and placed in the incubator for 45 minutes at 37°C. After that, Cells were washed two times with 1mL 1X PBS to remove ROCKi and cells bound nonspecifically to the PEGDA gel bed (Fig 1B(ii)).

**Differentiation.** After seeding, hESCs were treated by gastrulation assay previously described in [22], or by ectodermal differentiation assay as described in [7], or with reagents as described in the text.

### Immunostaining, fixed cell imaging and analyses

Immunostaining followed standard protocols as previously described in [21]. Primary and secondary antibodies were diluted in the blocking solution as described in [2,21]. Dilutions are listed in the reagents table (S1 Table). All immunostaining data were acquired by imaging fixed cell colonies on micropatterned dishes or slides using multi z-sections at 20X resolution NA 0.75 on an Olympus FV1200 laser scanning confocal microscope. All experiments were performed at least twice with consistent results. For images taken with multiple z-slices, background subtraction, maximum z projection and alignment were performed as described in [2].

**Living/Dead viability assay by calcein AM and propidium iodide.** Cell viability was assessed by calcein-AM (living) and propidium iodide (PI) (dead) double staining. The double staining solution was prepared with 2 μM calcein-AM and 4.5 μM PI in PBS. Seeding of ESI017 was performed as described above in the section *Cell seeding on micropatterned surface*. The cells were cultured in mTeSR1 medium with ROCKi at 37°C for 24 hours. Next, the staining solution was added into each well and incubated at 37°C for 15 minutes. After washing, the cell colonies on micropatterned slides were imaged using multi z-sections at 10X resolution NA 0.40 on an Olympus FV31S-SW laser scanning confocal microscope. Cell counting was performed by image analysis with FIJI [19] and ilastik [20], following the steps: making max intensity projection and splitting channels in FIJI, creating binary masks for the live cell and dead cell channels in ilastik, performing watershed and analyzing particles on the cell masks in FIJI. The viability was calculated as living cell count over total cell count (= living cell count + dead cell count). ESI017 cells growing on an LN521 coated glass surface were used as a positive control.

**Proliferation assay by Ki67 immunostaining.** Cell proliferation was assessed by Ki67 assay. After seeding, ESI017 were cultured in mTeSR1 medium with ROCKi at 37°C for 24 hours. Next, the cells were fixed and immunostained (with Ki67 and DAPI) following protocols as described above in section *Immunostaining, fixed cell imaging and analyses*. The cell colonies on micropatterned slides were imaged by using multi z-sections at 10X resolution NA 0.40 on an Olympus FV31S-SW laser scanning confocal microscope. A binary nuclear mask indicating nuclear positions was created by segmenting the DAPI channel by using ilastik [20] for each colony. Then the Ki67 channel intensity in each nucleus was calculated and plotted using MATLAB scripts. ESI017 cells growing on an LN521 coated glass surface were used as a positive control.

## Results

### Direct fabrication of PEGDA hydrogel on glass bottom dish

We used PEGDA to create a hydrogel bed with geometric features on the glass culture surface. The fabrication of PEGDA by photolithography was first described in [23]. Glass surfaces were functionalized by 3-(Trimethoxysilyl) propyl-methacrylate (TPM) prior to fabrication. Covalent bonds between the TPM and the acylate groups of the PEGDA molecules prevented

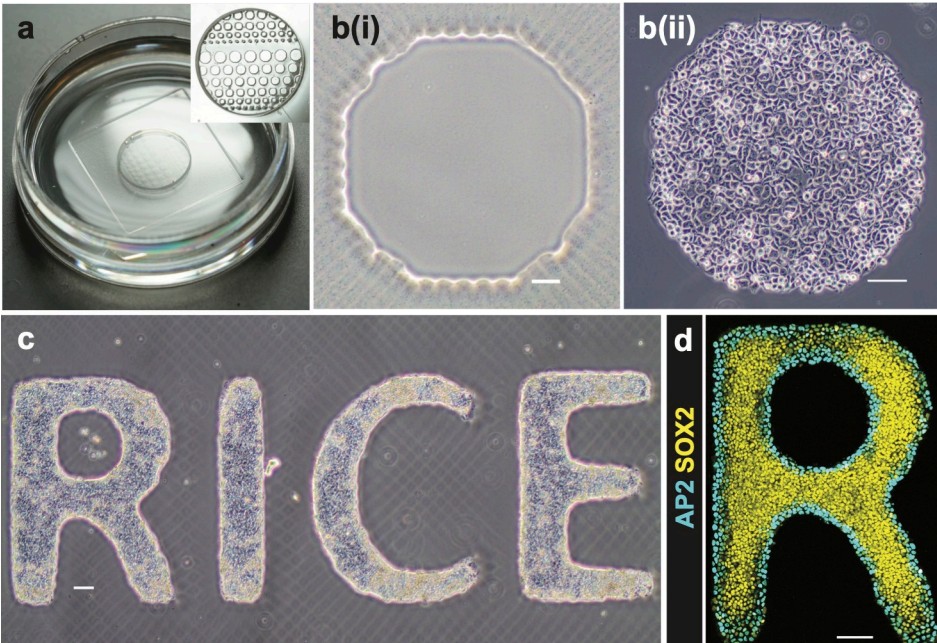

**Fig 2. hESC culture and induction on micropatterned PEGDA glass-bottomed dish.** (**a**) Photographs of a micropatterned glass-bottomed dish with PBS. (**b**) hESCs attached to micropatterned PEGDA coated glass surface: (i) bright field image of an octagon pattern (ii) hESCs attached well 45 min post cell seeding. Scale bar = 100 μm. (**c**) hESCs attached to "RICE" pattern were cultured for 3 days. Scale bar = 100 μm. (**d**) Immunostained hESC patterns for SOX2 and AP2 26 h post BMP treatment at 50 ng/mL. Scale bar = 100 μm. hESCs self-organized to form an outer ring of extra-embryonic cells (ISL1+), and the inner pluripotent cells (SOX2+).

detachment of the hydrogel bed from the glass surface during subsequent cell culture or sample processing. The PEGDA hydrogel was formed by free-radical polymerization of acrylate and methacrylate groups initiated by a blue light source (405 nm). Rather than physically manufacturing a new photomask for each pattern, the use of a digitally tunable virtual photomask provides rapid experimental iteration. The fabrication protocol on glass-bottom dishes is summarized in Fig 1A and described in Materials and Methods. Fig 2A shows photographs of micropatterned PEGDA hydrogel fabricated directly on the glass bottom of the petri dish. After fabrication, PEGDA hydrogels were immersed in PBS and swelled to their final size. We found that the micropatterned gels were stable and could be used for cell patterning experiments for at least 8 months after fabrication.

## Characterization of micropatterned PEGDA gel and cell pattern

Next, we sought to characterize the dimensions of the micropatterned PEGDA gel. First, we exposed the PEGDA gel on a glass surface to different blue light exposure times (4, 6, 8 and 10 s), and measured the thickness (Fig 3A). For each exposure time, three micropatterned PEGDA gels were fabricated; and measurements were performed on each gel at three different locations. As expected, the gel thickness linearly increases with the exposure time (Fig 3A). The thickness of the gel is substantially larger than the height of hESCs grown on a glass surface. Therefore, the micropatterned gel created a well-shaped space for cells to attach and grow.

Varying the blue light exposure times not only changed the gel thickness, but also affected the final sizes of patterns on the gel, compared to the pattern sizes on the photomask. We

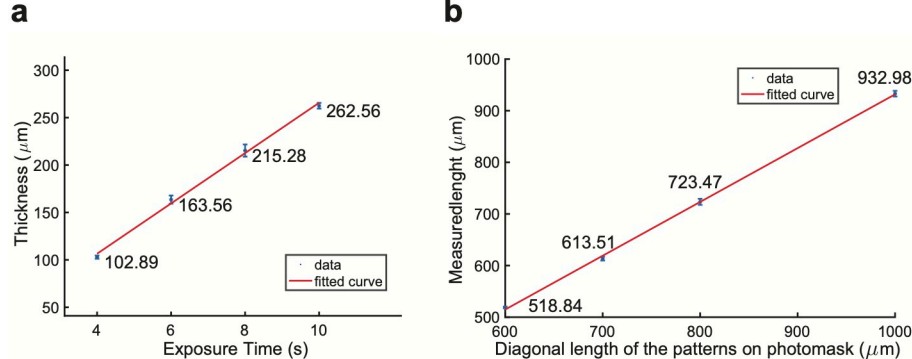

**Fig 3. Characterization of micropatterned PEGDA gel thickness and cell pattern size.** (**a**) Measurement of thickness of gel bed using different UV exposure time in fabrication. Unit: μm. The linear polynomial curve fitted to the data points has the equation of val(x) = p1*x + p2, where coefficients (with 95% confidence bounds) are p1 = 26.54 (21.93, 31.14), p2 = 0.317 (-33.53, 34.16). (**b**) Measurement of the sizes of circular cell patterns. All gels used were exposed to UV for 6 s. Major axis length of each cell colony was acquired, which represents the measured size of the cell pattern. Unit: μm. The linear polynomial curve fitted to the data has the same equation of val(x) = p1*x + p2, but with coefficients (with 95% confidence bounds) p1 = 1.042 (0.9707, 1.114) and p2 = -110.5 (-167, -54.1).

created circular patterns with diameters designed to be 600, 700, 800, and 1000 μm and then measured their actual sizes (Fig 3B). Each circular pattern was approximated with square pixels of size 50 μm. Cell colonies seeded onto the patterns have smoother edges than those of the gel patterns, as shown in (Fig 2B and compare to gel pictures in Fig 2A). All micropatterned gels were fabricated by using a UV exposure time of 6 seconds. For each pattern size, 5 patterns were measured following the methods described before. As shown in Fig 3B, measured lengths were slightly smaller than those on the photomask, and the measured and designed lengths were positively and linearly correlated with a slope close to 1.

## Cell attachment, growth, and proliferation on the micropatterned surfaces

We then investigated cell attachment and growth on the micropatterned surfaces. First, hESCs were seeded at a density of $2 \times 10^6$ cells/mL onto the LN521-coated surface. LN521 attached to the methacrylated glass surface but not the PEGDA after 2.5-hour incubation at 37°C. hESCs attached specifically to the LN521 coated glass surface and grew into colonies with the geometric restriction imposed by the micropatterned PEGDA gel bed (Fig 2B(ii)). Cells that settled on the PEGDA surface could be easily removed by washing with PBS, which did not affect the cell attachment to the LN521-coated glass surface. A more complicated "RICE" letter pattern was fabricated and seeded with hESCs (Fig 2C). On the third day, the cells remained firmly attached to the non-hydrogel areas and grew into uniform colonies with clear and smooth edges. We also tested mouse C2C12 cells on the micropatterned surface without LN521 and found good attachment and geometric patterning. Accordingly, multiple cell types can grow on micropatterned surfaces created with patterned PEGDA.

Next, cell viability and proliferation were also assessed. Live/dead double staining (using calcein-AM and PI) images of hESCs growing on micropatterned or the control glass surfaces are shown in Fig 4A. The viability of cells on the the micropattern group was slightly lower than the control group (84.0% and 85.9%, respectively), but this difference was not statically significant (Fig 4C). Fig 4B shows confocal images of hESCs immunostained for Ki67, a marker of proliferative cells. In both micropattern and control images, Ki67 expression was detected in every cell nucleus. Applying the same threshold to both distributions (see red lines in Fig 4D), the positive rate of micropattern and control groups were 98.2% and 99.3%,

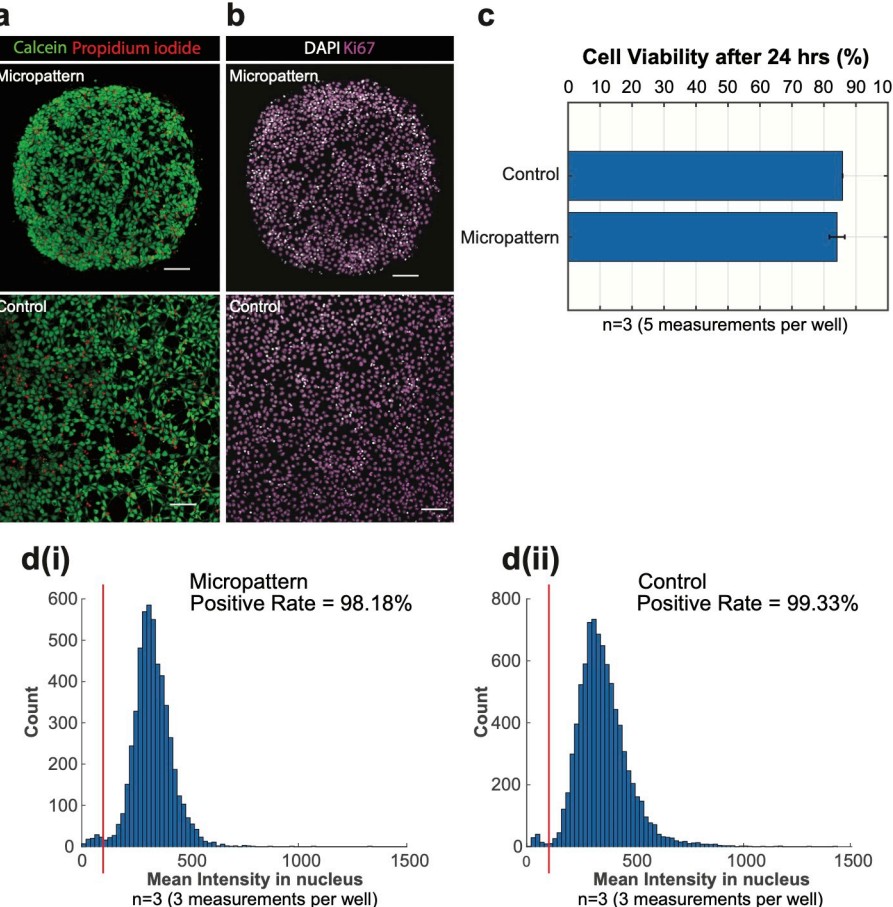

**Fig 4. Cell viability and proliferation assessments after 24 hours.** (**a**) Confocal images of live/dead staining of hESCs on LN521-coated micropatterned (top) or glass (bottom, control) surfaces with calcein-AM (live) and propidium iodide (dead) after 24-hour-culture. Scale bar = 100 μm. Colony size: Diameter of 650 μm. (**b**) Confocal images of hESCs immunostained for Ki67 on micropatterned (top) or bare glass (bottom, control) surfaces after 24-hour-culture. Scale bar = 100 μm. Colony size: Diameter of 650 μm. (**c**) The quantities of live and dead cells were quantified from the calcein-AM/PI staining images. Cell viability is calculated as a percentage of living cell (# of living cells/(# of living cells + # of dead cells). The results were reported as the mean ± standard deviation. P<0.05. (**d**) Cell proliferation assessed by Ki67 assay was quantified and analyzed from Ki67 immunostaining images, and plotted as histograms (micropatterns in (i), and control in (ii)). The red line in each plot indicated the threshold used to calculate the Ki67 positive rate.

respectively (Fig 4D(i) and 4D(ii)). After growing for 24 hours, the cell density in the colony center on the micropattern was similar to the cell density on the control surface, while the colony edge had a higher density of cells, as expected for cells in confined culture. These differences in cell density may explain minor differences between cell viability and proliferation in the different conditions. Overall, cells grew similarly on glass surfaces with micropatterned gel beds and control surfaces without micropatterning.

## Self-organized patterning of hESCs on the micropatterned surfaces

We next investigated whether these methods were compatible with previously described protocols for self-organized patterning of hESCs on micropatterned surfaces. Warmflash et al developed a method utilizing hESCs on micropatterns, which recapitulates some aspects of gastrulation [2], and it has subsequently been used to study many aspects of signaling and

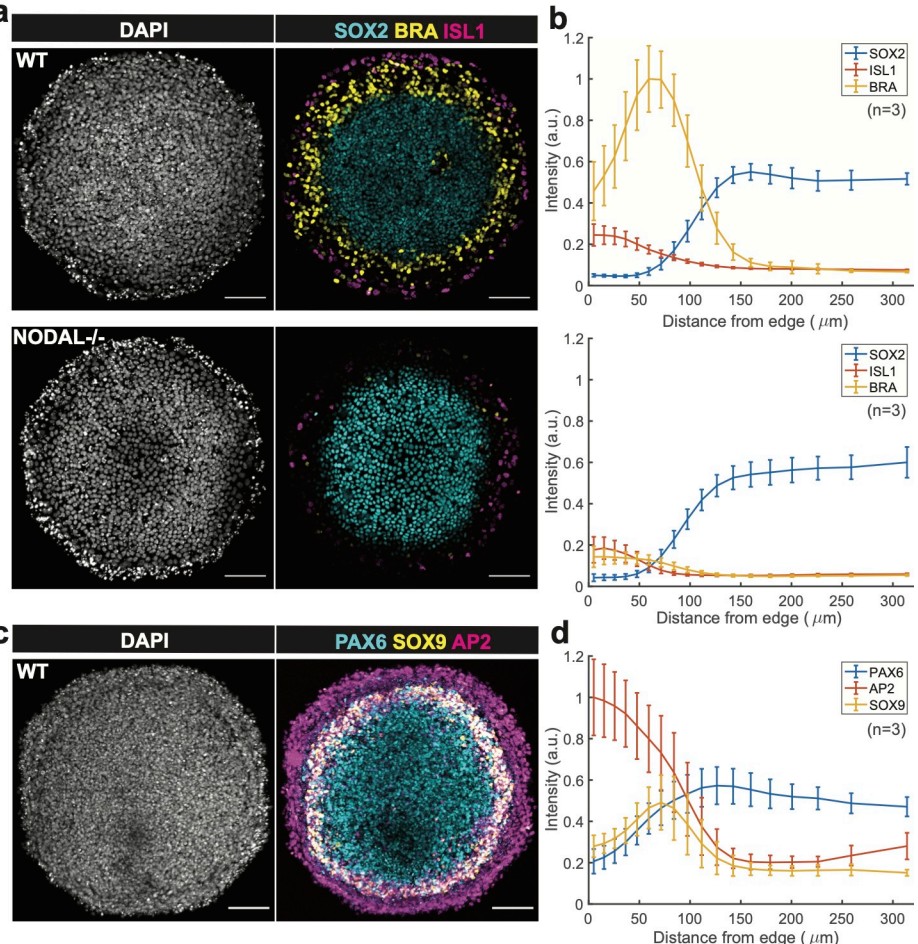

**Fig 5. Reproduction of gastrulation and neural ectodermal induction results with the PEGDA-based micropatterning method.** (**a**) Confocal images of cell colonies immunostained for SOX2, BRA, and ISL1 at 48 h post BMP treatment in different conditions: Control and NODAL −/− cells were treated with 50 ng/ml BMP4. BMP, Bone Morphogenic Protein. Scale bar = 100 μm. Colony size: Diameter of 650 μm. (**b**) SOX2, BRA, and ISL1 levels of the wild type group (top) and Nodal -/- group (bottom) were quantified as a function of distance from the colony center (n = 3). (**c**) Confocal images of cell colony immunostained for PAX6, SOX9, and AP2. Patterns were initially treated for 3 days in N2B27 media with 10 ng/mL SB and then subsequently induced for 1 day in N2B27 media with SB, BMP4 and IWP2. Scale bar = 100 μm. Colony size: Diameter of 650 μm. (**d**) PAX6, SOX9, and AP2 levels of the ectodermally induced group were quantified as a function of distance from colony center (n = 3).

differentiation [1–10]. Most of these studies use commercially available slides (CYTOO). In this model, spatial patterns of cells with four lineages (an outer ring of extra-embryonic cells, a pluripotent or ectodermal center, and rings mesoderm and endoderm in between) can be obtained by a simple treatment with 50 ng/mL BMP4 of hESCs for 2 days. In comparison to wild-type hESCs, NODAL knockout cells (*NODAL -/-*) had less mesoderm differentiation due to the absence of functional NODAL protein [6]. To test whether these results can be recapitulated using the patterning method devised here, wild-type and *NODAL -/-* hESCs were seeded on the micropatterned surfaces, and ROCKi was removed 45 mins after seeding. On day 2, the 48-hr treatment was initiated for both cell lines with 50ng/mL BMP4 in the media. As shown in Fig 5A, wild-type cell colonies had an outer ring of ISL1+ extra-embryonic cells, an inner ring of BRA+ mesoderm/primitive streak, and a SOX2+ pluripotent center. On the other hand, *NODAL -/-* cell colony has reduced BRA expression on the inner ring, while SOX2

+ ISL1+ expression was not altered. In short, the results by using PEGDA micropatterned surface is consistent with those using commercial micropatterned slides [6].

We also tested whether this method could be applied to a model of human ectoderm patterning described previously [7]. This is a more challenging method as it requires 6 days of growth on the micropatterned surface. Britton et al established a two-phase protocol that could differentiate spatial patterns of the four major ectodermal fates (neural plate, neural crest, placode, and surface ectoderm) along the radial axis of the colony. The protocol is composed of 3 days in N2B27 media supplemented with SB431542 (10 μM) (called ectoderm induction media), and another 2 days in ectoderm induction media with BMP4 (5 ng/mL) and with IWP2 (4 μM) on the fifth day only. The cells were kept in mTeSR1 with SB431542 for one night after seeding which we found to improve cell adhesion, and then the ectodermal patterning protocol described above was performed. Morphologically, the resulting colonies had a multilayered structure with increased density of cells in a ring-shaped region with reproducible radius. The colonies consisted of an AP2α+ outer ring of surface ectodermal lineage, a PAX6+ center of neural plate, and a ring of neural crest with co-expression of PAX6 and SOX9 (Fig 5B). Thus, the results on PEGDA micropatterned surfaces and using commercial slides [7] are consistent in both morphology and fate patterning.

## Discussion and conclusion

We have described an efficient and highly adaptable fabrication method for micropatterning glass surfaces with PEGDA using the fabrication apparatus, SLATE [15]. After blue light (405 nm) exposure with a virtual photomask, a micropatterned hydrogel is firmly attached to the glass surface in a particular pattern. The PEGDA prevents adhesions of cells, while excellent cell attachment, spatial patterning, viability and proliferation were seen with cells growing in the regions without hydrogels. Our results with hESCs indicate the feasibility of performing developmental studies on these micropatterned surfaces. The method can be used in any format of culture dish with a glass bottom–we successfully used it in 35 mm glass-bottomed dishes, and 18-well and 8-well glass-bottomed plates.

PEGDA was selected for its mild crosslinking conditions, good stability, biocompatibility, and stiffness, and its ability to prevent cell adhesion. First, the crosslinking of PEGDA occurs at room temperature with a photo-initiator (LAP) and safe blue (405 nm) light source, which are mild conditions that can be easily achieved. Second, the crosslinking bond PEGDA forms as an ester, which typically remain stable in aqueous environments for several weeks to months. If longer culture periods are needed, the PEGDA could be substituted with PEG diacrylamide, which has an aqueous half-life on the order of years. In our hands, the micropatterned dishes were still good for use after 8 months when stored at 4°C. Third, in the previous work with the SLATE system [15], PEGDA hydrogels were shown to have more than 80% water content, which were able to support high human mesenchymal stem cell viability and osteogenic activity. Fourth, PEGDA serves as a cell-repellant substrate here, because the polymer is entirely comprised of short repeating C-C-O units, which lack both peptide and non-specific cell-adhesive moieties. Since PEGDA is an extremely hydrophilic polymer, there is little opportunity for the hydrophobic effect to drive protein adsorption, which could otherwise aid in cell adhesion if a hydrophobic substrate was chosen.

Human LN-521 is a cell adhesion protein expressed in the inner cell mass of the blastocyst staged human embryo, along with other member of the laminin family [24]. Recombinant LN-521 coating in hESC culture provides high survival and growth rate, allowing long-term self-renewal and pluripotency maintenance [25]. We have previously shown that LN-521 yields

better cell attachment and patterning results compared with Matrigel [22]. Thus, LN-521 was used throughout this study.

Traditional photolithography can also be used to generate hydrogel micropatterns. In either traditional photolithography or SLATE, a spatial light modifier is needed to generate a 2D pattern of light. Traditional photolithography employs a physical photomask to spatially control these light patterns, physically manufactured as an opaque sheet with 2D patterns of holes or transparencies. Every time the pattern is changed, a new photomask must be created. The complete and commercially available system, SLATE, which uses a standard projection system with dynamically tunable photomasks, allows patterns to be rapidly optimized with any fabrication. Thus, compared to other micropatterning methods described in the introduction, our method is cost-effective and simple, with the ability to rapidly change the patterns by simply changing the virtual photomask on a computer. The ability to fabricate in any format allows the use of more cost-effective formats with lower volumes of reagents needed during cell culturing and sample processing. Itoga et al developed a maskless photolithographic method with modified liquid crystal display projector for cellular micropatterning in a two-step process: fabrication of positive-type photoresist micropatterns on the silanized coverslips, and then polymerization of the polyacrylamide (PAAm) layer on photoresist-free area. After the photoresist was removed with acetone, a PAAm micropatterned layer on the coverslip surface was ready for cell culture. However, the thermally initiated polymerization need to be as long as 6 h at 40˚C [26]. Recently, Yang et al also developed a UV-curing method for PEGDA hydrogel microstructure fabrication [27]. This protocol is similar to the one described here with differences in the UV wavelength and PEGDA molecular weight. They showed the ability to created cell chains with L929 cells. Here we showed that a similar method can be used to culture more biologically relevant hESCs and to perform developmental patterning studies.

Our method is currently restricted by the spatial resolution of the blue light projector, which has 50 μm pixels. Consequently, circular patterns with diameters below approximately 600um appeared as octagons. However, the resolution is only limited by the quality of the projector and could be improved. Another limitation is that the hydrogel thickness can only be controlled by adjusting exposure time to UV, which also affects the actual pattern size. One way to make hydrogel with a thin and controlled thickness is to place a coverslip on top of the pre-polymer solution that is added onto the glass surface. The pre-polymer solution then spreads out and forms a thin layer of liquid under the coverage of the coverslip. The less solution that is added, the thinner the hydrogel is after UV exposure. In summary, our simple and efficient method provides an alternative solution for fabrication of micropatterns for cell culture and spatial patterning hESC studies and can be improved upon in the future to improve the resolution in pixel size and thickness.

## Supporting information

**S1 Table. Key resource table.**
(DOCX)

## Acknowledgments

We thank George Britton for helpful discussions on ectodermal patterning.

## Author Contributions

**Conceptualization:** Ye Zhu, Jordan S. Miller, Aryeh Warmflash.

**Funding acquisition:** Aryeh Warmflash.

**Investigation:** Ye Zhu.

**Methodology:** Ye Zhu, Daniel Sazer.

**Project administration:** Aryeh Warmflash.

**Supervision:** Jordan S. Miller, Aryeh Warmflash.

**Visualization:** Ye Zhu.

**Writing – original draft:** Ye Zhu, Aryeh Warmflash.

**Writing – review & editing:** Ye Zhu, Daniel Sazer, Jordan S. Miller, Aryeh Warmflash.

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
