## [Decision Letter · Decision Letter 0]

1 Feb 2021

PONE-D-21-00568

Rapid fabrication of hydrogel micropatterns by projection stereolithography for studying self-organized developmental patterning

PLOS ONE

Dear Dr. Warmflash,

Thank you for submitting your manuscript to PLOS ONE. After careful consideration, we feel that it has merit but does not fully meet PLOS ONE’s publication criteria as it currently stands. Therefore, we invite you to submit a revised version of the manuscript that addresses the points raised during the review process.

We look forward to receiving your revised manuscript.

Kind regards,

Yi Cao

Academic Editor

PLOS ONE

Journal Requirements:

3. Please upload a copy of Supporting Information Table 1 which you refer to in your text on page 16.

Reviewers' comments:

Reviewer's Responses to Questions

**Comments to the Author**

1. Is the manuscript technically sound, and do the data support the conclusions?

Reviewer #1: Yes

Reviewer #2: Yes

Reviewer #3: Yes

Reviewer #4: Yes

Reviewer #5: Yes

2. Has the statistical analysis been performed appropriately and rigorously? 

Reviewer #1: Yes

Reviewer #2: Yes

Reviewer #3: Yes

Reviewer #4: Yes

Reviewer #5: Yes

3. Have the authors made all data underlying the findings in their manuscript fully available?

Reviewer #1: Yes

Reviewer #2: Yes

Reviewer #3: Yes

Reviewer #4: Yes

Reviewer #5: Yes

4. Is the manuscript presented in an intelligible fashion and written in standard English?

Reviewer #1: Yes

Reviewer #2: No

Reviewer #3: Yes

Reviewer #4: Yes

Reviewer #5: Yes

5. Review Comments to the Author

Reviewer #1: Zhu et al. report a gel patterning method together with protein coating creates a surface suitable for human embryonic stem cell (hESC) attachment and growth with minimal non-specific adhesion. The manuscript further demonstrates the self-patterning of hESCs based on the previous gastrulation and ectodermal induction protocols. The results are impressive, except for the lack of solid quantification data. I suggest a major revision of this manuscript.

Some major concerns :

1) The Measurement of gel thickness is demonstrated in figure 3a, but why not show the SEM or bright filed image of the increasing thickness of the patterned gel? e.g., labeling the thickness on each image, and the curve size is none necessary to be so large to cover so much space in one figure.

2) Only the staining results are demonstrated in the results of hESCs culturing, but the cell viability and proliferation rate need to be tested and quantified, which are significant results to demonstrate the biocompatibility of the material and techniques.

3) Reproduction of gastrulation and neural ectodermal induction results with the micropatterning method was demonstrated in figure 4. However, it is better to do the marker staining quantification via image analysis, and the marker protein staining results of NODAL-/- in figure 4b is missing, which should also be provided.

Reviewer #2: In this work, the authors report Rapid fabrication of hydrogel micropatterns by projection stereolithography for studying self-organized developmental patterning，Self-organized patterning of mammalian embryonic stem cells on micropatterned surfaces has previously been established as an in vitro platform for early mammalian

developmental studies, complimentary to in vivo studies. Traditional micropatterning methods, such as micro-contact printing, involve relatively complicated fabrication procedures, which restricts widespread adoption by biologists. Here, we demonstrate a rapid method of micropatterning by printing hydrogel micro-features onto a glassbottomed culture vessel. The micro-features are printed using a projection stereolithography bioprinter yielding hydrogel structures that geometrically restrict the attachment of cells or protein. Compared to traditional, physical photomasks, a digitally tunable virtual photomask is used in the projector to generate blue light patterns that enable rapid iteration with minimal cost and effort. We show that a protocol that makes use of this method together with LN521 coating creates a surface suitable for human embryonic stem cell (hESC) attachment and growth with minimal non-specific adhesion. We further demonstrate that self-patterning of hESCs following previously published gastrulation and ectodermal induction protocols achieves results comparable with those obtained with commercially available plates. Overall the work quality is large and is meaningful, the work deserves to be published. However some points should be further clarified, therefore major revisions and re-valuation may be considered:

1. Some grammar and format errors should be carefully checked through the manuscript.

2. In the abstract part, the application of this method together with LN521 should be further introduced in more detail.

3. In the introduction part, this technique has been used by previous scientists, why it can be used to two-dimensional (2D) micropatterning of hydrogel features on glass surfaces using a single virtual photomask, the idea of using the technique is still not especially clear.

4. In the introduction part, if the Methacrylation of Glass Surface is different or similar to prevous method, if is so, please give the previous literature.

5. As for the Characterization of micropatterned PEGDA gel and cell pattern, as for the expression “The thickness of the gel is substantially larger than the height of hESCs grown on a glass surface. Therefore, the micropatterned gel created a well-shaped space for cells to attach and grow.” some recent literatures such as , Biomaterials, 2021, 265,120456, Mol. Pharmaceutics 2020, 17, 4, 1300-1309. Chin Chem Lett 2020, 31(12), 3178-3182, ACS Nano, 2020, 14 (10), 13536-13547 should be added.

6. The stability of gel and cell pattern should be considered.

7. Discussion and Conclusion should be divided into two parts, the discussion is not enough. The comparsivon between the current result and previous work is needed.

8. The reference style should be uniformed.

Reviewer #3: In this manuscript, the authors illustrated a rapid micropatterning procedure to induce self-organized mammalian embryonic stem cells in vitro. The microscopic features were fabricated using a stereolithography bioprinter, generating hydrogel structures that restrict cell attachment in spatial distribution. Compared with traditional micropatterning methods, the procedures digitally adjusted photomasks that minimized cost and enabled rapid iteration. The authors illustrated that this micropatterning procedure could induce the differentiation of human embryonic stem cells into gastrula and ectodermal cells and achieved comparable results to commercially available plates. However, there are still a few issues to be clarified prior to the publication of this manuscript.

1. Why did the authors choose poly(ethylene glycol) diacrylate (PEGDA) as the material of hydrogel?

2. What were the advantages of stereolithography compared to traditional micropatterning methods?

3. The surface morphology of micropatterned surfaces should be characterized by a scanning electron microscope.

4. Why did cells not attach to the surface of PEGDA hydrogel?

5. What was the mechanism of cell attachment to the LN521-coated glass surface?

6. What were the application prospects of micropatterning surfaces?

7. The recently published review and research articles should be discussed in the revision, for example, Nature Methods 2020, 17 (1), 50-54; Journal of Tissue Engineering 2020, 11, 2041731420943839; Cells 2019, 8 (8), 886.

Reviewer #4: The authors investigated a rapid method of micropatterning by printing hydrogel micro-features onto a glass-bottomed culture vessel. They printed the micro-features using a projection stereolithography bioprinter yielding hydrogel structures that geometrically restrict the attachment of cells or protein. The authors claimed that self-patterning of human embryonic stem cells following previously published gastrulation and ectodermal induction protocols achieves results comparable with those obtained with commercially available plates. This is an interesting study. However, it still needs some minor revisions before publication.

1. The abstract needs to rewrite. The authors didn’t highlight enough their own discoveries. In another word, they discussed too much about the previous stuff, which could reduce the significance of this study.

2. Where is the data for characterization of hydrogels? I couldn’t get a general idea bout the hydrogel.

3. The authors claimed that the hydrogel thickness is the limitation in this study. However, there are also some other concerns. For example, the stiffness of the hydrogel may be important for this study, which could affect a lot of stem cell differentiation. Moreover, why the authors only use the blue light projector? The properties of the hydrogel could be tuned with different light sources. Therefore, I suggest the authors try different methods or do more characterizations for the hydrogel following the literature below. At least, the authors need to give some discussion about those questions.

(1) DOI: 10.1021/acsami.5b11811

(2) DOI: 10.1016/j.cclet.2018.06.009

Reviewer #5: Self-organized patterning of mammalian embryonic stem cells on micropatterned surfaces has previously been established as an in vitro platform for early mammalian developmental studies, complimentary to in vivo studies. Therefore, the work is interesting and impportant. The authors have proved their idea and thus the manuscript can be published as it is.

6. PLOS authors have the option to publish the peer review history of their article (what does this mean?). If published, this will include your full peer review and any attached files.

Reviewer #1: No

Reviewer #2: No

Reviewer #3: No

Reviewer #4: No

Reviewer #5: No

---

## [Author Response · Author response to Decision Letter 0]

27 Apr 2021

Please see response to reviewers document uploaded with the manuscript

---

## [Decision Letter · Decision Letter 1]

30 Apr 2021

Rapid fabrication of hydrogel micropatterns by projection stereolithography for studying self-organized developmental patterning

PONE-D-21-00568R1

Dear Dr. Warmflash,

We’re pleased to inform you that your manuscript has been judged scientifically suitable for publication and will be formally accepted for publication once it meets all outstanding technical requirements.

Kind regards,

Yi Cao

Academic Editor

PLOS ONE

Additional Editor Comments (optional):

Reviewers' comments:

Reviewer's Responses to Questions

**Comments to the Author**

1. If the authors have adequately addressed your comments raised in a previous round of review and you feel that this manuscript is now acceptable for publication, you may indicate that here to bypass the “Comments to the Author” section, enter your conflict of interest statement in the “Confidential to Editor” section, and submit your "Accept" recommendation.

Reviewer #1: All comments have been addressed

Reviewer #3: All comments have been addressed

Reviewer #4: All comments have been addressed

2. Is the manuscript technically sound, and do the data support the conclusions?

Reviewer #1: Yes

Reviewer #3: Yes

Reviewer #4: Yes

3. Has the statistical analysis been performed appropriately and rigorously? 

Reviewer #1: Yes

Reviewer #3: Yes

Reviewer #4: Yes

4. Have the authors made all data underlying the findings in their manuscript fully available?

Reviewer #1: Yes

Reviewer #3: Yes

Reviewer #4: Yes

5. Is the manuscript presented in an intelligible fashion and written in standard English?

Reviewer #1: Yes

Reviewer #3: Yes

Reviewer #4: Yes

6. Review Comments to the Author

Reviewer #1: (No Response)

Reviewer #3: All my previous comments have been well addressed. The manuscript has been well revised and is valuable to be accepted in the current form.

Reviewer #4: The authors have revised the manuscript accordingly, I think this study is ready for publicaiton in the current format.

7. PLOS authors have the option to publish the peer review history of their article (what does this mean?). If published, this will include your full peer review and any attached files.

Reviewer #1: No

Reviewer #3: No

Reviewer #4: No

---

## [Editor Report · Acceptance letter]

20 May 2021

PONE-D-21-00568R1 

Rapid fabrication of hydrogel micropatterns by projection stereolithography for studying self-organized developmental patterning 

Dear Dr. Warmflash:

I'm pleased to inform you that your manuscript has been deemed suitable for publication in PLOS ONE. Congratulations! Your manuscript is now with our production department. 

Kind regards, 

on behalf of

Dr. Yi Cao 

Academic Editor

PLOS ONE